# Simulation of Work Hardening in Machining Inconel 718 with Multiscale Grain Size

**DOI:** 10.3390/ma16093562

**Published:** 2023-05-06

**Authors:** Kejia Zhuang, Zhuo Wang, Linli Zou, Changni Fu, Jian Weng

**Affiliations:** 1Hubei Digital Manufacturing Key Laboratory, School of Mechanical and Electronic Engineering, Wuhan University of Technology, Wuhan 430070, China; 2State Key Laboratory of Intelligent Manufacturing Equipment and Technology, Huazhong University of Science and Technology, Wuhan 430074, China; 3Xiamen Golden Egret Special Alloy Corporation Limited, Xiamen 361000, China

**Keywords:** work hardening, Inconel 718, finite element simulation, grain size

## Abstract

Machining nickel-based alloys always exhibits significant work-hardening behavior, which may help to improve the part quality by building a hardened layer on the surface, while also causing severe tool wear during machining. Hence, modeling the work-hardening phenomenon plays a critical role in the evaluation of tool wear and part quality. This paper aims to propose a numerical model to estimate the work-hardening layer for a deeper understanding of this behavior, employing both recrystallization-based and dislocation-based models to cover workpieces with multiscale grain sizes. Different user routines are implemented in the finite element method to simulate the increase in hardness in the deformed area due to recrystallization or changes in the dislocation density. The validation of the proposed model is performed with both literature and experimental data for Inconel 718 with small or large grain sizes. It is found that the recrystallization-based model is more suitable for predicting the work-hardening behavior of small-grain-size Inconel 718 and the dislocation-based model is better for that of large-grain-size Inconel 718. Further, as an important type of cutting tool in machining Inconel 718, the chamfered tools with different edge geometries are employed in the simulations of machining-induced work hardening. The results illustrate that the uncut chip thickness and chamfer angle have a significant influence on the work-hardening behavior.

## 1. Introduction

Inconel 718 is an essential superalloy, commonly utilized in the aerospace industry (e.g., gas turbines), as well as in the shipbuilding, petrochemical, and nuclear industries due to its exceptional corrosion resistance and the ability to maintain its physical properties even under high temperatures [1,2,3]. Nevertheless, the machinability of Inconel 718 is poor due to severe work-hardening behavior during the cutting process [4]. A certain degree of work hardening (DWH) will significantly enhance the strength and wear resistance of the workpiece surface, thereby improving the performance of the workpiece. However, excessive work hardening can lead to drastic changes in the metal’s microstructure, increasing both the cutting force and wear on the tool, which makes continuous cutting challenging. Therefore, studying the phenomenon of work hardening is imperative.

Numerous experimental studies have been conducted to examine the influence of cutting parameters, tool geometries, and cutting conditions on work hardening from a macroscopic point of view. According to research conducted by Hua and Liu [5], a higher cutting speed leads to a higher DWH, with only a slight difference in the depth of the hardening layer. A greater feed rate leads to a higher DWH and a deeper hardening layer. Umbrello [6] discovered that increasing the cutting speed and feed rate resulted in a higher surface hardness and deeper hardening layer on the workpiece, accompanied by a noticeable refinement of the grain structure. Dinesh et al. [7] conducted a study on the effects of cutting speed, feed rate, and cutting depth on the DWH and found that the lowest DWH occurs under conditions of a low cutting speed and feed rate. Pawade [8] utilized Taguchi orthogonal design to thoroughly investigate the influence of the cutting parameter and tool geometry on work-hardening behavior in the cutting process of Inconel 718. The research indicated that both the tool geometry and cutting depth of the tool had the most significant influence on DWH. The depth of the hardening layer was found to be greatest when using a tool with a chamfer angle (i.e., the angle between the chamfer and the tool rake face plane). However, increasing the chamfer angle resulted in a decrease in the depth of the hardening layer.

It is widely recognized that microstructure differences between the machined surface of the workpiece and its subsurface are caused by different machining parameters. Numerical methods are widely employed to simulate this behavior because of their high prediction accuracy and great ability to capture intricate in-process physical variables. Two primary theories of simulation studies on work hardening are recrystallization and dislocation density. For the recrystallization theory, Jafarian et al. [9,10] proposed a model based on the hardness of the material, combined with the model based on the Zener–Hollomon parameter (Z–H model) to study the influence of tool micro-edges including the edge radius and chamfer angle for Inconel 718 microstructure evolution during the cutting process, especially the effect of grain size and hardness. Caruso et al. [11] studied the microstructure evolution of the SAE 8620 cutting process by combining the hardness-based Johnson–Cook (J–C) material constitutive model and the Z–H model. The results indicate that the average prediction error for grain size is approximately 20%, while for surface hardness, it is 4%. Arisoy et al. [12,13] combined the improved temperature-based flow softening material model and the three-dimensional finite element (FE) model and used the recrystallization model based on Johnson–Mehl–Avrami–Kolmogorov (JMAK) to study the microstructure changes of the machined surfaces of Inconel 100 alloy and titanium alloy Ti6Al4V, including the changes in phase distribution, volume fraction, and hardness, during the cutting process. Xu et al. [14] predicted the grain size and work hardening of Ti6Al4V during the cutting process by using the recrystallization model based on JMAK, and studied the grain refinement and work hardening caused by dynamic recrystallization through the microstructure evolution of titanium alloy. When it comes to the dislocation density theory, Ding et al. [15] used the dislocation density-based model (DDB model) to study the cutting speed, feed rate, and tool rake angle on dislocation density, grain size, hardness, and metamorphic layer thickness of Al6061–T6 alloy. A low cutting speed, high feed rate, and negative tool rake angle contributed to a higher dislocation density, thinner deformation layer, and smaller grain size, resulting in improved hardness of the machined surface. Li et al. [16] used the DDB model to study the effect of the cutting speed, feed rate, and radial cutting depth on the grain size evolution of the H13 steel surface during the milling process. With the increase in the cutting speed, feed rate, and radial cutting depth, the grain size of the machined surface changes slightly and gradually returns to its initial size. Liu et al. [17] simulated the cutting process of OFHC copper using the DDB model and concluded from the experiment and simulation results that the fluctuation of a cutting force at high cutting speed was caused by the evolution and distribution of the dislocation density. The dislocation density on the machined surface increased with the increase in cutting speed, while it decreased in the chips.

The different grain sizes of materials under the same processing conditions will show differences in the cutting force, surface hardness, and surface integrity. Komatsu et al. [18,19] researched the effect of stainless-steel grain size on the micro-milling process. The study found that the formation of burrs can be controlled by reducing the grain size. Compared with ordinary grain steel, when cutting ultra-fine-grain steel, the shear force is small, and the surface finish is markedly improved. The findings of Olovsjö et al. [20] demonstrated that large burrs only occurred in large grain size materials of the wrought Alloy 718 during cutting. Meanwhile, the depth of the deformation layer of the large-grain-size material was greater than the small-grain-size material. Yang et al. [21] studied the effect of grain size on the surface characteristics of pure iron during orthogonal cutting and found that the residual stress on the machined surface decreased with an increase in grain size. In addition, the hardness of small-grain-size material is greater than that of large-grain-size material. Wu et al. [22] studied the influence of grain size on cutting force and burr formation in the copper micro-cutting process and found that small grain size is beneficial for reducing cutting force and burrs. In the study of the process of tool wear, Polvorosa et al. [23] found that the small grain size of the material resulted in larger flank wear during the milling of nickel-based superalloys. However, few researchers have studied the influence of grain size on work-hardening behavior in the cutting process.

In this paper, an FE model of orthogonal cutting is established. Moreover, the Z–H model based on recrystallization theory and the DDB model based on dislocation theory are combined to systematically simulate the work-hardening behavior of Inconel 718 with different grain sizes and discuss the applicability of different models. Finally, a suitable model for large-grain-size Inconel 718 is used to study the effects of uncut chip thickness and tool edge geometries (the chamfer angle and chamfer length) on work-hardening indicators (DWH, surface hardness, and hardening layer depth).

## 2. Overview of the Work

This paper focuses on the simulations of the work-hardening phenomenon when cutting Inconel 718 with multiscale grain size, and the overview is shown in Figure 1. First, multiple cutting simulations are performed based on DEFORM. Then, two different models (Z–H and DDB models) are employed to predict the hardness in the subsurface. The model constants will be calibrated until the maximum prediction error is obtained. For Inconel 718 with a large grain size, orthogonal cutting experiments are conducted, and the hardness in the subsurface area is measured by nanoindentation. Further, the microstructure images are taken for model validation. For Inconel 718 with a small grain size, literature data from reference [6] are used to compare the performance of the two models. Then, the predicted results given by the Z–H and DDB models are compared, and their performances are discussed. Finally, the effect of the uncut thickness and tool edge geometries (the chamfer angle and chamfer length) on the work-hardening behavior of Inconel 718 with a large grain size is further investigated using the DDB model.

## 3. Simulation and Experimentation

### 3.1. FE Models of the Orthogonal Process

DEFORM^TM^ v11.0 developed by Scientific Forming Technologies Corporation (SFTC, Columbus, OH, USA) was employed to simulate the orthogonal cutting process of Inconel 718. The plane strain coupling thermos-mechanical analysis was carried out under the orthogonal assumption. During the simulation, the updated Lagrange code and remeshing technique were adopted to realize the thermo-steady-state and mechanical-steady-state conditions. Very fine elements were defined near the cutting region (1 μm) to obtain more precise results and better chip geometry shape.

The mesh and boundary conditions of the FE model are shown in Figure 2. The tool was deprived of all displacements, and the workpiece could move only in the X direction. The bottom and left of the workpiece and the top and right of the tool were set to room temperature (*T_room_*). The top and right of the workpiece and the bottom and left of the tool (marked in red) were allowed to exchange heat with the environment. During the cutting process, the heat generated in the cutting zone was transferred to the workpiece, tool, chips, and environment [24]. A very high heat transfer coefficient of 10^5^ kW/(m^2^K) [25] was selected between the chip, tool, and workpiece so that the temperature field could reach a stable state in a short time. The hybrid friction model regarding the chip–tool and tool–workpiece interface was considered. Specifically, the shear friction factor was employed in the sticking contact area, and the coulomb friction factor was employed in the sliding area. In this model, the shear friction factor and coulomb friction factor were set to 1 and 0.3, respectively [25].

In this paper, Cockroft and Latham’s damage criterion [26] was considered to simulate the formation of serrated chips during orthogonal cutting. The criterion is expressed as:(1)$$\int_{0}^{\varepsilon_{f}}\sigma_{1}d\varepsilon =D$$
where *ε_f_* is the effective strain, *σ*_1_ is the maximum principal stress, and *D* is the critical damage value. When the integral of the largest tensile principal stress of any element reaches the *D* value, the fracture will occur, and chip segmentation will start. In this model, *D* = 120.

The flow stress behavior was defined using the J–C material constitutive model, which describes the plastic deformation of materials at different ranges of strain, strain rate, and temperature. The model is illustrated as follows:(2)$$\sigma =\left(A+B\varepsilon^{n}\right)\left(1+C\left(\ln\frac{\dot{\varepsilon }}{{\dot{\varepsilon }}_{0}}\right)\right)\left(1-{\left(\frac{T-T_{room}}{T_{melt}-T_{room}}\right)}^{m}\right)$$
where *σ* is the flow stress, *ε* is the plastic strain, $\dot{\varepsilon }$ is the strain rate, ${\dot{\varepsilon }}_{0}$ is the reference strain rate, *T* (°C) is the workpiece temperature, *T_room_* is the room temperature (20 °C), and *T_melt_* is the melting temperature of the material (1300 °C for Inconel 718). *A*, *B*, *C*, *m*, and *n* are J–C material constitutive model parameters, where *A* (MPa) is the yield strength, *B* (MPa) is the hardening modulus, *C* is the strain rate sensitivity coefficient, *m* is the thermal softening coefficient, and *n* is the hardening coefficient. The J–C material constitutive model relevant parameters are listed in Table 1.

### 3.2. FE Models of Microstructure Changes

#### 3.2.1. The Z–H Model

The FE simulation software DEFORM^TM^ v11.0 can simulate the macroscopic deformation and microstructure evolution of the metal-forming process and provide the secondary development subroutine interface. A user subroutine for the Z–H model was implemented in this software to predict microstructure changes such as recrystallization grain size and hardness. In this paper, the Zener–Hollomon equation (Equations (3) and (4) is employed to predict recrystallization [28], and the Hall–Petch equation (Equation (5)) is employed to calculate the hardness change [29]. A more detailed description of these equations is as follows:(3)$$Z=\dot{\varepsilon }\exp\left(\frac{Q}{RT}\right)$$
(4)$$d=bZ^{m}$$
(5)$$HV=C_{0}+C_{1}d^{-0.5}$$
where *Z* is the Z–H parameter, $\dot{\varepsilon }$ is the strain rate, *Q* is the apparent activation energy (368.5 kJ/mol) [25], *R* is the universal gas constant (8.3145 J/K·mol) [25], *T* is the temperature (K), *d* is the recrystallized grain size, and *HV* is the value of the Vickers hardness. In addition, *b*, *m*, *C*_0_, and *C*_1_ are material constants, which will be given in Section 3.2.3. When the strain generated on the material surface during cutting exceeds the critical strain, the recrystallization process begins. The critical strain *ε_cr_* was first proposed by Sui et al. [30] (for the hot continuous rolling process of Inconel 718 alloy), and then Jafarian et al. [25] calibrated the formula for the cutting process by adding a constant “*c*”. The value of critical strain *ε_cr_* is related to strain rate $\dot{\varepsilon }$ and temperature *T*. The final critical strain *ε_cr_* is represented by the following formula:(6)$$\varepsilon_{cr}=\left(0.00234{\dot{\varepsilon }}^{0.1293}\exp\left(\frac{5729.863}{T+273}\right)\right)/c$$
where *ε_cr_* is the critical strain, $\dot{\varepsilon }$ is the strain rate, *T* is the temperature (°C), and *c* is the calibration constant.

The implemented strategy for the prediction of hardness variation during the orthogonal cutting process of Inconel 718 is shown in Figure 3. In each time step of the simulation, the critical strain *ε_cr_* of all elements is calculated and compared with the corresponding strain obtained from the FE simulation. When the strain exceeds the critical strain *ε_cr_*, recrystallization occurs, and the hardness changes. As a result, the hardness value is updated. Otherwise, the hardness will not change.

#### 3.2.2. The DDB Model

A user subroutine for the DDB model is implemented in DEFORM^TM^ v11.0 software to predict microstructure evolution. Estrin et al. [31] first introduced the model to study the grain refinement of copper during isotropic corner extrusion, and then Ding et al. [32,33] applied the model to predict the evolution of the material dislocation density, chip formation, and grain refinement in the cutting process for the titanium alloy. In this model, the dislocations are generated by the plastic deformation of the workpiece, and there are two types of dislocation densities, namely cell interior dislocation density (*ρ_c_*) and cell wall dislocation density (*ρ_w_*), which can be further subdivided into geometrically necessary dislocation density (*ρ_wg_*) and statistical dislocation density (*ρ_ws_*). The relevant dislocation density evolution rate equations are as follows:(7)$${\dot{\rho }}_{c}=\alpha^{*}\frac{1}{\sqrt{3}b}\sqrt{\rho_{w}}{\dot{\gamma }}_{w}^{r}-\beta^{*}\frac{6}{bd{\left(1-f\right)}^{1/3}}{\dot{\gamma }}_{c}^{r}-k_{0}{\left(\frac{{\dot{\gamma }}_{c}^{r}}{{\dot{\gamma }}_{0}^{r}}\right)}^{-1/n}\rho_{c}{\dot{\gamma }}_{c}^{r}$$
(8)$${\dot{\rho }}_{ws}=\beta^{*}\frac{\sqrt{3}(1-f)}{fb}\sqrt{\rho_{w}}{\dot{\gamma }}_{c}^{r}+(1-\xi )\beta^{*}\frac{6{\left(1-f\right)}^{2/3}}{bfd}{\dot{\gamma }}_{c}^{r}-k_{0}{\left(\frac{{\dot{\gamma }}_{w}^{r}}{{\dot{\gamma }}_{0}^{r}}\right)}^{-1/n}\rho_{ws}{\dot{\gamma }}_{w}^{r}$$
(9)$${\dot{\rho }}_{wg}=\xi \beta^{*}\frac{6{\left(1-f\right)}^{2/3}}{bfd}{\dot{\gamma }}_{c}^{r}$$
(10)$${\dot{\rho }}_{w}={\dot{\rho }}_{ws}+{\dot{\rho }}_{wg}=\beta^{*}\frac{\sqrt{3}(1-f)}{fb}\sqrt{\rho_{w}}{\dot{\gamma }}_{c}^{r}+\beta^{*}\frac{6{\left(1-f\right)}^{2/3}}{bfd}{\dot{\gamma }}_{c}^{r}-k_{0}{\left(\frac{{\dot{\gamma }}_{w}^{r}}{{\dot{\gamma }}_{0}^{r}}\right)}^{-1/n}\rho_{w}{\dot{\gamma }}_{w}^{r}$$

The three terms on the right side of Equations (7) and (10) represent the contributions of different mechanisms to the dislocation density. The first one is the dislocation growth caused by the Feank–Read dislocation source, the second one is the dislocation transformation from interior dislocations to cell wall dislocations due to plastic deformation during the processing, and the third one is the dislocation annihilation caused by thermal dynamic reversion at high temperatures. Parameter *ξ* is the proportion of geometrically necessary dislocation density in the cell wall dislocation density, and 𝑏 is the Burgers vector, ${\dot{\gamma }}_{0}^{r}$ is the reference shear strain rate. *α^*^*, *β^*^*, and *k*_0_ are parameters to control the evolution rate of dislocation density, *n* is the parameter of temperature sensitivity, *f* is the volume fraction of the cell wall dislocation density, and the parameters of *n*, *f*, and *d* are reported in the following equations:(11)$$k_{0}=0.016T+0.4$$
(12)n=B/T
(13)$$f=f_{\infty }+(f_{0}-f_{\infty })e^{-\gamma^{r}/{\tilde{\gamma }}^{r}}$$
where *B* is a constant, *T* is the temperature (K), $\gamma^{r}$ is the shear strain, ${\tilde{\gamma }}^{r}$ is the reference shear strain, and *f*_0_ and *f_∞_* are the proportions of the initial and final dislocation density in the total dislocation density, respectively.

Parameters ${\dot{\gamma }}_{w}^{r}$ and ${\dot{\gamma }}_{c}^{r}$ are the shear strain rates of the cell wall and cell interior, respectively. In order to satisfy the strain compatibility, the following equations are used to describe the shear strain rates:(14)$${\dot{\gamma }}_{w}^{r}={\dot{\gamma }}_{c}^{r}={\dot{\gamma }}^{r}$$
(15)$${\dot{\gamma }}^{r}=M\dot{\varepsilon }$$
where *M* is the Taylor factor and $\dot{\varepsilon }$ is the equivalent strain rate.

The final grain size *d* is related to the total dislocation density *ρ_tot_*.
(16)$$\rho_{tot}=f\rho_{w}+(1-f)\rho_{c}$$
(17)$$d=K/\sqrt{\rho_{tot}}$$
where *K* is the material coefficient. The strengthening of material hardness depends on the dislocation density due to severe plastic deformation (SPD). The relationship between the hardness (Δ*HV_SPD_*) of the material and the dislocation can be added to the current model, and the equation of hardness (Δ*HV_SPD_*) strengthening [34] is as follows:(18)$$\Delta HV_{SPD}=k_{h}MaGb\sqrt{\rho_{tot}}$$
where *k_h_* and *a* are constants. *M* is the Taylor factor, *G* is the shear modulus, and *b* is the Burgers vector. In this paper, a constant *µ* is defined as:(19)$$\mu =k_{h}MaGb$$

The flow chart of the implemented strategy for the prediction of hardness variation during the orthogonal cutting process of Inconel 718 is shown in Figure 4. At each step of the simulation, the current time is compared to the predetermined total simulation time. If the current time exceeds the set total simulation time, the subroutine will stop executing the calculation and deliver the result; otherwise, the subroutine will continue to execute.

#### 3.2.3. Models Calibration

To ensure accurate predictions, a calibration phase is required to define material constants using the experimental results. Therefore, conducting a sensitivity analysis to understand the influence of individual parameters on the hardness is crucial.

For the Z–H model, FE simulation analysis reveals that the surface hardness and DWH vary significantly when parameter *b* is changed; however, the change in the work-hardening layer depth is not significant. Parameter *c* primarily influences the work-hardening layer depth. To begin the calibration, the same value utilized by Jafarian [25] is chosen as the starting value for the sensitivity analysis since these values are good predictors of Inconel 718 surface hardness. Parameters *m*, *C*_0_, and *C*_1_ are set at 0.01, 378, and 298.4, respectively, and *b* is a function of uncut chip thickness, as follows:(20)$$b=600h^{2}-139.8h+10.2$$

Subsequently, calibration is conducted to acquire the suitable value for parameter *c*. For different initial grain sizes of Inconel 718, parameter *c* is set to different values: 100 and 1000 for the small grain size (18 μm) and large grain size (115 μm), respectively.

For the test condition of *h* = 0.05 mm, the calibration results of hardness variation near the machined surface can be seen in Figure 5. In the machined surface and subsurface, the hardness has a significant change. The maximum hardness is in the range of 10~20 μm from the machined surface, which is not only affected by the mechanical load but also by the thermal load to a substantial extent. Along the depth direction from the machined surface, the influence of thermal load gradually decreases, but the hardness in the range of 50 μm from the machined surface is also significantly larger than the bulk hardness. The hardness variation curve at the Q point of the machined surface along the depth direction decreases rapidly from the machined surface to 80 μm and begins to converge at approximately 115 μm, which is in good agreement with the experimentally measured hardening layer depth of 104.5 μm [6,25].

A comparison between the predicted and experimental results for the three test conditions (*h* = 0.05, 0.075, and 0.1 mm) is shown in Figure 6. Very good agreement is observed between the predicted and experimental results for the hardness of the machined surfaces for all test conditions. In terms of the depth of the hardening layer, the average error is 5.17%, and the simulation result is generally good, although under some test conditions (1 out of 3 experiments), the prediction result is higher than the corresponding experimental result (13.6% error). The overall average error (Figure 7) of the machined surface hardness and depth of the hardening layer fluctuates between 2.05% and 7.9%.

The DDB model is adapted to the cutting process of the Inconel 718 by modifying the important parameters of the model.

The microstructure pictures (Figure 8) of the cross-section view of the workpiece are obtained via Scanning Electron Microscopy (SEM) (Zeiss, Germany), using test 7 (cutting speed *v_c_* = 50 m/min, uncut chip thickness *h* = 0.08 mm, chamfer length *L* = 100 μm, chamfer angle *θ* = 15°) as an example. The results indicate that slip lines can be observed in the subsurface area within the range of 20 µm under the machined surface, which means an increase in the dislocation density.

Under test 7 conditions, the predicted dislocation density, grain size, and surface hardness are shown in Figure 9. The prediction results show that there is an obvious deformation layer along the depth direction from the machined surface, which is consistent with the SEM pictures (Figure 8). The dislocation density and hardness are the highest at the machined surface and gradually decrease along the depth direction of the machined surface until they are equal to the bulk material. The grain size variation trend is opposite to the hardness variation.

In the paper by Hu et al. [35], the DDB model parameters for the shot-peened process of Inconel 718 are given. On this basis, some of the parameters are modified for the cutting process of Inconel 718 in this paper.

The specific parameters for the model are listed in Table 2. Specifically, the parameter *K* and the previously defined constant *µ* have different values for different grain sizes of Inconel 718, and for different initial grain sizes of Inconel 718, parameters *K* and *µ* are set to different values. For small-grain-size Inconel 718, the parameters *K* and *µ* are set to 119 and 0.0153, respectively. For large-grain-size Inconel 718, parameters *K* and *µ* are set to 769 and 0.03, respectively.

### 3.3. Experimentation

The accuracy of the outcomes obtained by the FE method heavily depends on the accuracy of the input values after calibration. Thus, to obtain an accurate prediction of work-hardening phenomena, a suitable experimental scheme needs to be devised.

Experimental tests for the large-grain-size Inconel 718 under dry cutting conditions on a CAK5058nzj CNC lathe were performed as shown in Figure 10. The uncoated carbide triangle blades with a chamfer angle are employed to provide a 0° rake angle and a 6° clearance angle in the orthogonal cutting experiment. The blade is mounted on a Sandvik shank. The workpiece dimensions are 50 mm × 25 mm × 3 mm (length × width × height) (Figure 10c) and the material is the large-grain-size Inconel 718 with an average Vickers hardness of 345 and an average grain size of 115 μm. The related parameters of the cutting experiment are listed in Table 3 test 1~9. After processing, the workpiece is cut into 10 mm × 10 mm × 3 mm (length × width × height) small samples. The sample surface is made smooth by grinding and polishing operations and then preserved by pressing the samples into the toner using a hot press inlay machine. The NanoTest Vantage4 is employed for nanoindentation measurements with a Berkovich indenter, applying a fixed maximum load of 50 mN in this study. We measured 75 points in total along the depth direction from the machined surface, with a spacing of 20 μm between every individual indentation, and 5 indentations are set for a specific depth value, as shown in Figure 10a,b.

For small-grain-size Inconel 718, testing conditions are conducted at the same cutting speeds (*v_c_* = 70 m/min) but different uncut chip thicknesses (*h* = 0.05, 0.075, and 0.1 mm). These are literature data from D. Umbrello [6]. The workpiece material has an average hardness value of 429 HV and an average grain size of 18 μm.

## 4. Results and Discussions

### 4.1. Two Different Models Predict Results

The Z–H model and DDB model are employed to predict the work-hardening behavior of Inconel 718, considering the small grain size (18 μm) and large grain size (115 μm).

For small-grain-size Inconel 718, FE simulations are conducted under two models for three experimental conditions (*h* = 0.05, 0.075, and 0.1 mm), respectively, and the fitting results of the simulated result and experimental data are shown in Figure 11.

The simulation results of both the Z–H model and DDB model can fit the experimental data of small-grain-size Inconel 718 (Figure 12), which indicates that the microstructure models based on recrystallization and dislocation can be used to explain the work-hardening phenomenon of small-grain-size Inconel 718. However, the Z–H model, based on recrystallization, is more suitable for small-grain-size Inconel 718.

For the large-grain-size Inconel 718, FE simulations were performed under four test conditions (tests 1, 2, 4, and 7) in Table 3 using the Z–H model and DDB model, respectively. The fitting of simulation results and experimental data for each of the four test conditions is shown in Figure 12.

The fitting of the predicted results and experimental data of the two models under the conditions of tests 1, 2, 4, and 7 is shown in Figure 12. It is apparent that the DDB model better fits the predicted results and experimental values. As can be seen under the optical microscope in Figure 13, grain refinement does not occur on either the machined surface or the subsurface, which indicates that recrystallization does not occur on the machined surface of the large-grain-size Inconel 718. The SEM pictures (Figure 8) show severe plastic deformation and obvious slip lines, which are reflections of dislocation densities. This further supports the conclusion that the DDB model, based on dislocation theory, is more suitable than the Z–H model, based on recrystallization theory, to explain the work-hardening phenomenon of large-grain-size Inconel 718.

After determining that the DDB model can better predict the large-grain-size Inconel 718, the effects of the uncut chip thickness, chamfer length, and chamfer angle on work-hardening behavior are studied by using the DDB model under different test conditions (test 1~9).

### 4.2. Analysis of Work-Hardening Behavior

Under different cutting parameters, tool chamfer angles, and chamfer lengths, the variation trends of work-hardening indices (DWH, surface hardness, and hardening layer depth) are shown in Figure 14.

The work-hardening characteristics of Inconel 718 are a limiting factor with poor machinability. As a result, the DWH needs to be controlled in the cutting of Inconel 718. Hence, in this study, the DWH calculated by Equation (21) illustrates the severity of the work-hardening behavior of the machined material. Equation (21) is described as follows [8,36]:(21)$$\mathrm{DWH}(\%)=\frac{HV_{s}-HV_{0}}{HV_{0}}\cdot 100\%$$
where *HV*_s_ and *HV*_0_ represent the hardness of surface and bulk material, respectively.

For the DWH, the maximum value is obtained when the large uncut chip thickness (0.08 mm) and small chamfer angle (15°) are used, and the minimum value is acquired when the small uncut chip thickness (0.04 mm) and small chamfer angle (15°) are used. The DWH is positively correlated with the chamfer angle and uncut chip thickness. When the large chamfer angle (25°) is used, the DWH is negatively correlated with the chamfer length. When the small chamfer angle (15°) is used, the DWH increases at first and then decreases with the increase in the chamfer length. The result of variance analysis shows that the uncut chip thickness and chamfer angle have a significant influence on the DWH, with the uncut chip thickness exerting the most significant influence, followed by the chamfer angle.

For the surface hardness, its changing trend is consistent with the DWH. The most significant impact on the surface hardness is the uncut chip thickness, followed by the chamfer angle. A larger uncut chip thickness leads to a greater cutting force and more severe plastic deformation on the machined surface, resulting in a larger plowing area under the cutting edge and aggravating the work hardening.

For the hardening layer depth, the maximum value is obtained when a small uncut chip thickness (0.04 mm) and large chamfer angle (25°) are used, and the minimum value is acquired when a large uncut chip thickness (0.08 mm) and small chamfer angle (15°) are used. The hardening layer depth is negatively correlated with the uncut chip thickness and positively correlated with the chamfer angle. With a small uncut chip thickness (0.04 mm), the hardening layer depth is negatively correlated with the chamfer length. With a large uncut chip thickness (0.08 mm), the hardening layer depth is positively correlated with the chamfer length. The result of variance analysis shows that the uncut chip thickness has a significant influence on the depth of the hardening layer.

## 5. Conclusions

In this paper, the Z–H model and DDB model are used to predict the work hardening behavior of Inconel 718 with large or small grain sizes, respectively, during the orthogonal cutting process, and the applicability of these two models to workpieces of different grain sizes is discussed. Based on the DDB model, the effects of the uncut chip thickness, chamfer angle, and chamfer length on the work-hardening behavior of large-grain-size Inconel 718 are also discussed. The main contributions can be drawn as follows:

(1)For the small-grain-size Inconel 718, the fitting effect of the Z–H model (i.e., recrystallization-based model) is better than the DDB model. For the large-grain-size Inconel 718, the predicted results of the DDB model are in better agreement with the experimental results.(2)The SEM pictures of the experiment show that there are slip lines in the surface area, but no obvious grain refinement is found. This indicates that the recrystallization-based model of work hardening is not suitable for the workpiece with a large-grain-size Inconel 718.(3)The DWH and the surface hardness are significantly affected by the uncut chip thickness and chamfer angle, among which uncut chip thickness is the most significant factor. With the increase in both the uncut chip thickness and chamfer angle, the surface hardness and DWH increase.(4)For the depth of the hardening layer, the uncut chip thickness has a significant influence. The depth of the hardening layer will decrease with increasing uncut chip thickness. In general, the uncut chip thickness has the most significant influence on the hardening behavior, followed by the chamfer angle and chamfer length.

In this study, the uncut chip thickness is relatively low (0.05, 0.075, and 0.1 mm). More work should be performed in the future under the conditions of a large uncut chip thickness where the work-hardening phenomenon is more significant.

## Figures and Tables

**Figure 1 materials-16-03562-f001:**
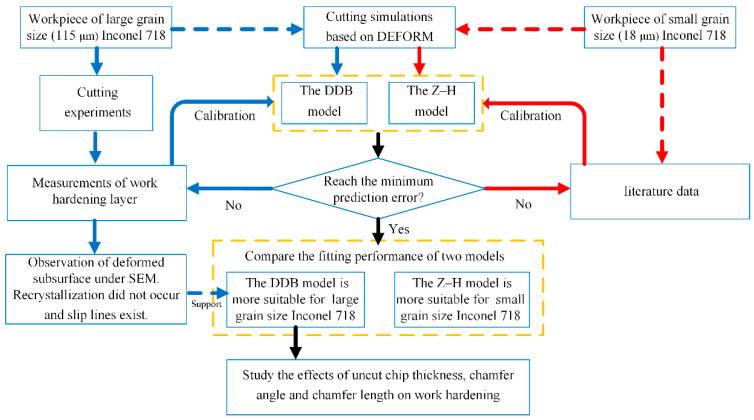
Overview of this study.

**Figure 2 materials-16-03562-f002:**
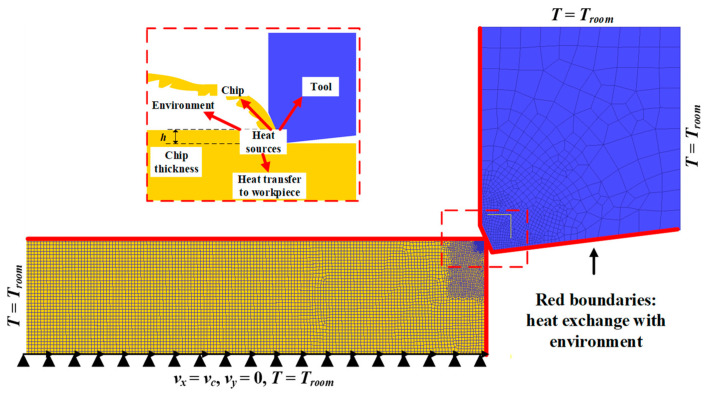
Mesh and boundary conditions for the FE model.

**Figure 3 materials-16-03562-f003:**
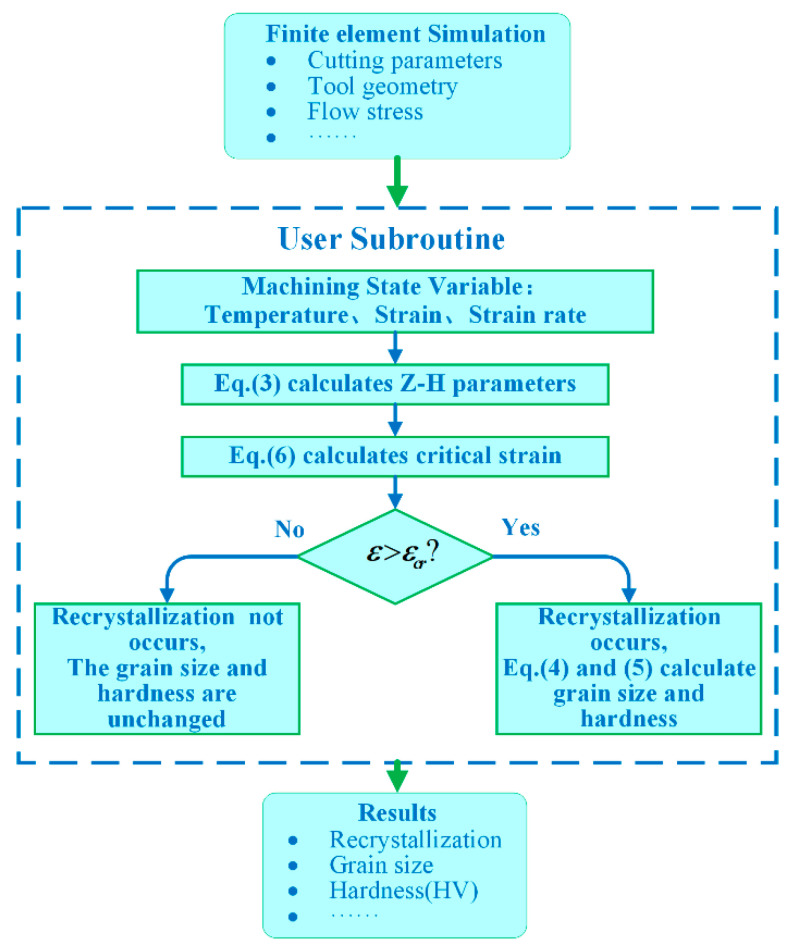
Flow chart of Z–H model subroutine.

**Figure 4 materials-16-03562-f004:**
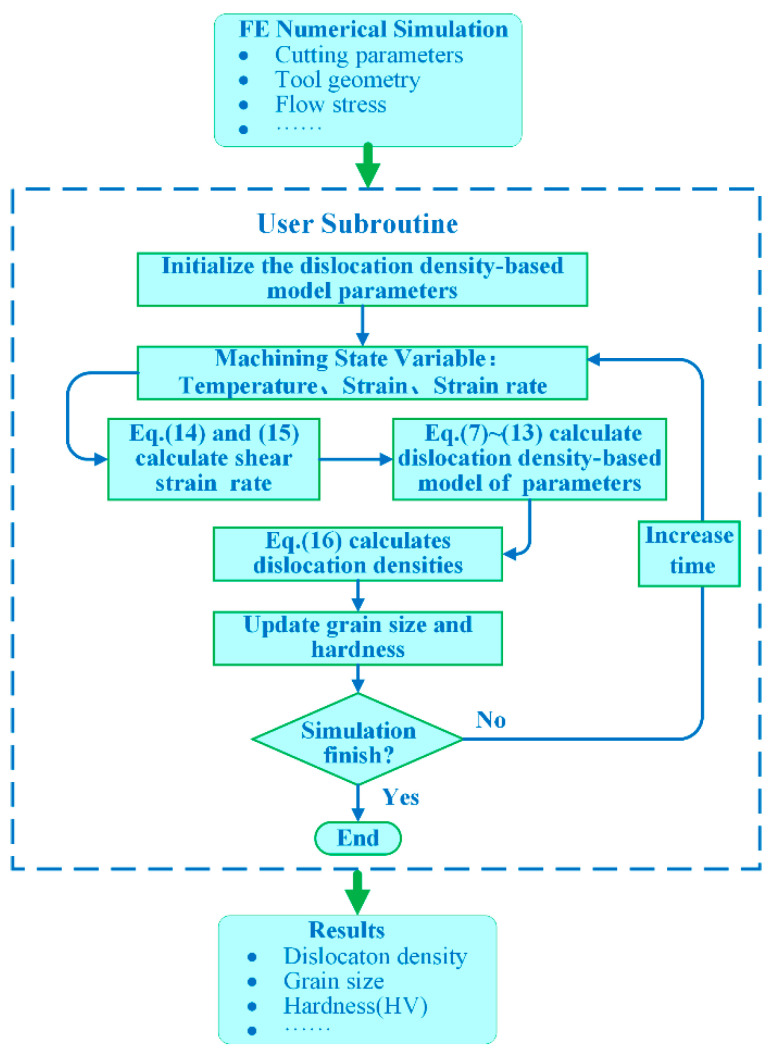
Flow chart of DDB model subroutine.

**Figure 5 materials-16-03562-f005:**
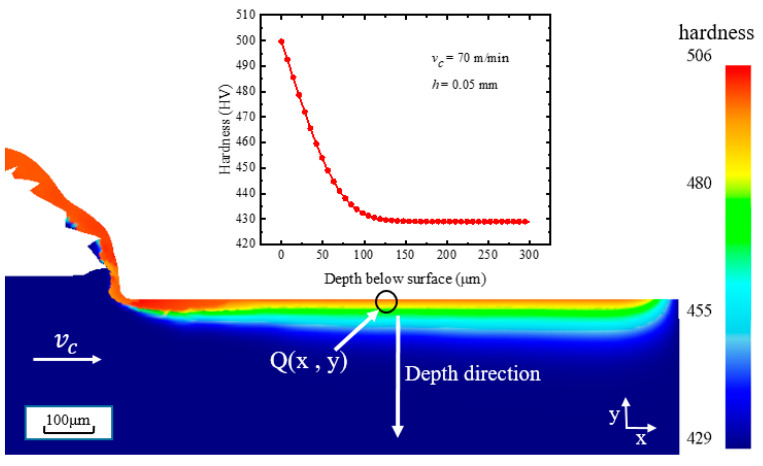
Simulated result of Z–H model under test condition *h* = 0.05 mm.

**Figure 6 materials-16-03562-f006:**
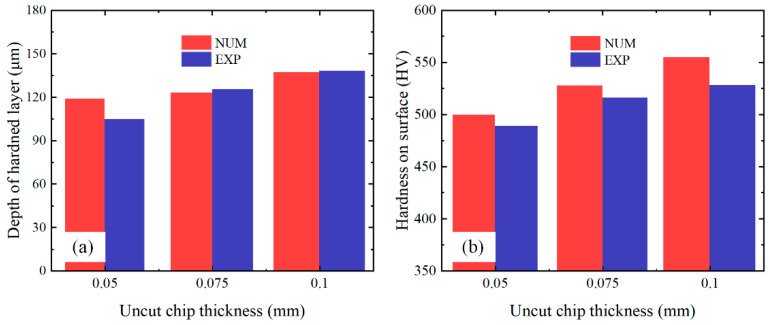
Comparison between experimental and predicted results of hardness variation depth (**a**) and surface hardness (**b**).

**Figure 7 materials-16-03562-f007:**
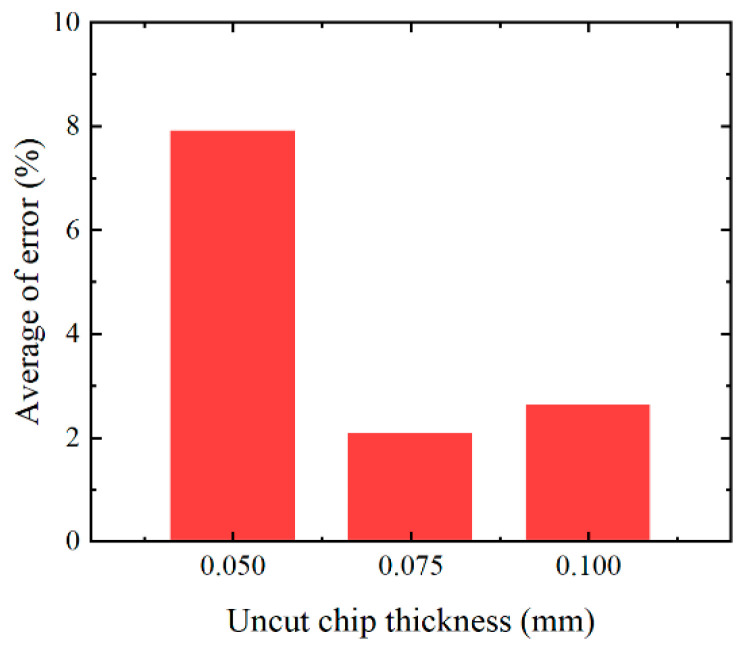
Overall average error for machined surface hardness and hardening layer depth.

**Figure 8 materials-16-03562-f008:**
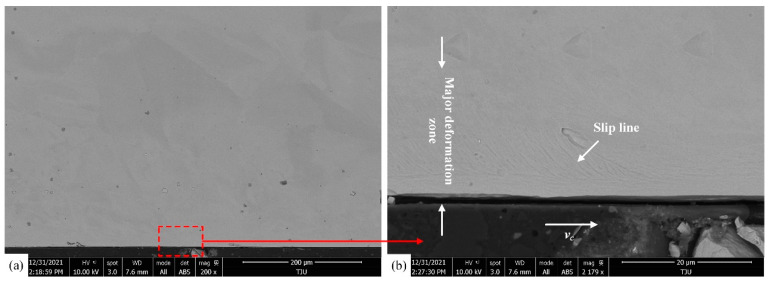
SEM of surface and subsurface layer (**a**) and subsurface deformation after machining at *v_c_* = 50 m/min, *h* = 0.08 mm, *L* = 100 μm, and *θ* = 15° (**b**).

**Figure 9 materials-16-03562-f009:**
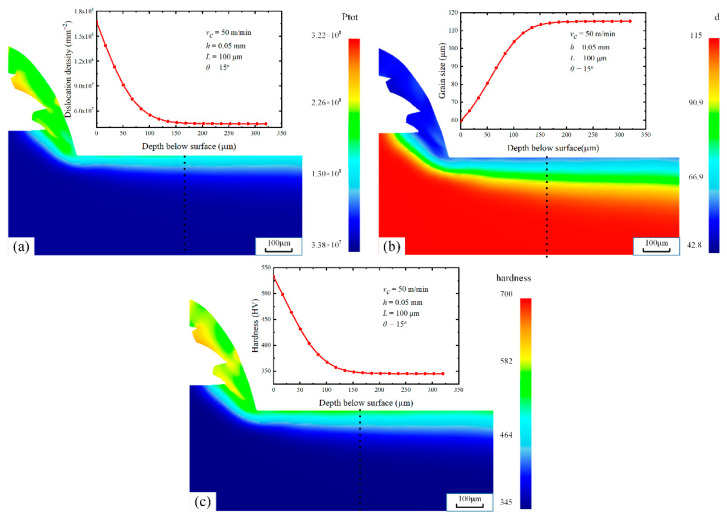
Simulated results of DDB model calibrated under test 7 condition. Dislocation density (**a**), grain size (**b**), and surface hardness (**c**).

**Figure 10 materials-16-03562-f010:**
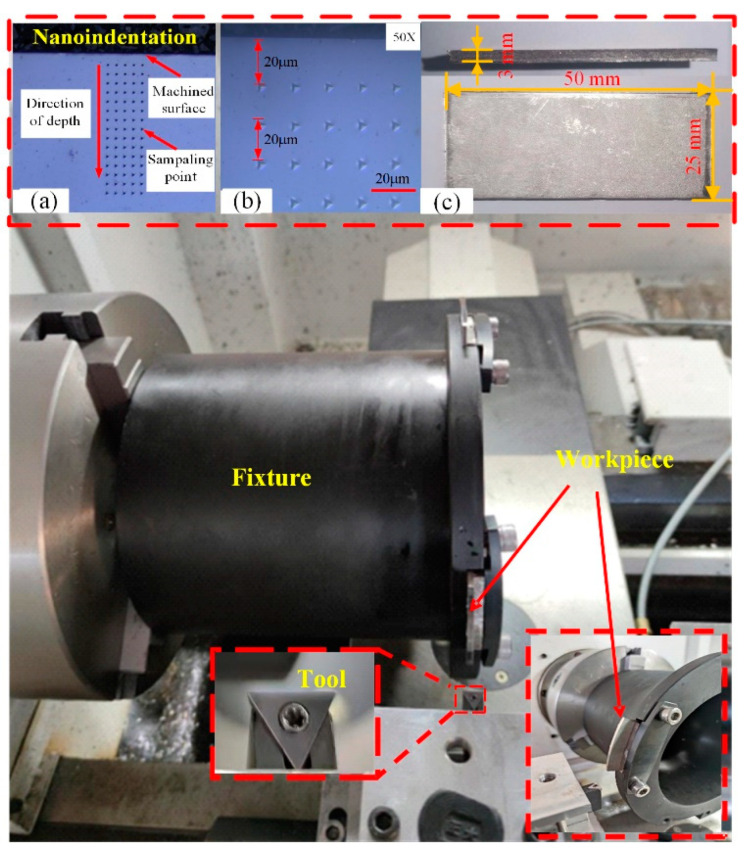
Settings of cutting experiments and hardness measurements. The nanoindentation measurement (**a**,**b**) and the workpiece dimensions (**c**).

**Figure 11 materials-16-03562-f011:**
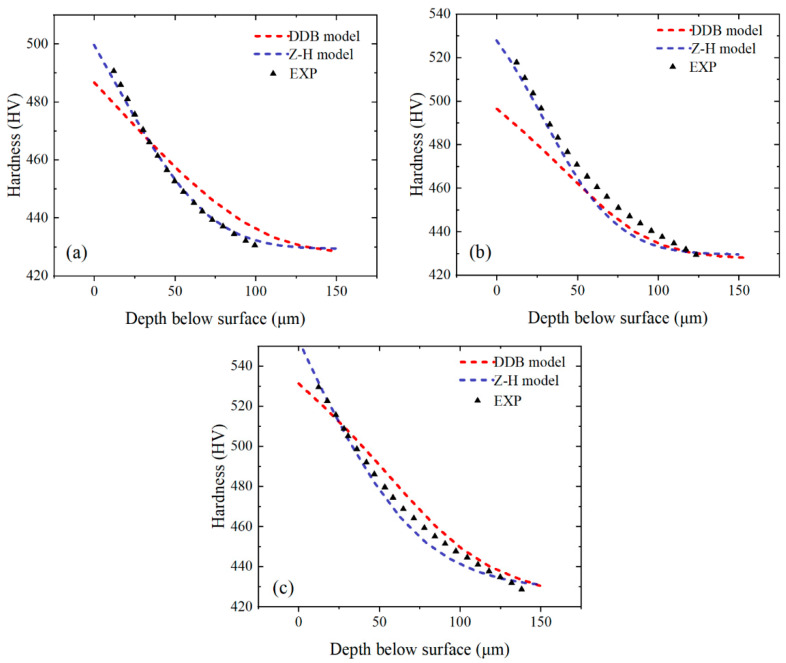
Comparison of the performance of DDB and Z–H models for Inconel 718 with small grain size based on literature data [6]. *h* = 0.05 (**a**), *h* = 0.075 (**b**), and *h* = 0.1 mm (**c**).

**Figure 12 materials-16-03562-f012:**
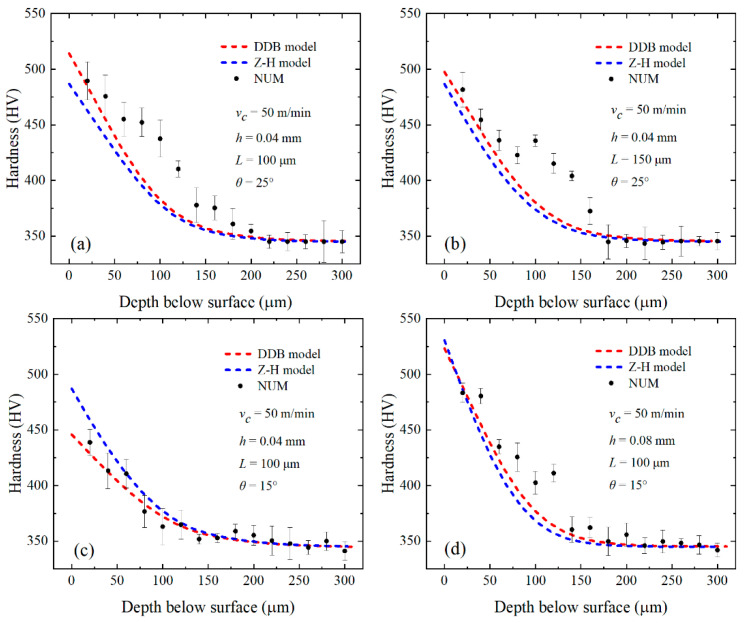
Comparison of the performance of DDB and Z–H models for Inconel 718 with large grain size based on experiments, see Table 3. Tests 1 (**a**), 2 (**b**), 4 (**c**), and 7 (**d**).

**Figure 13 materials-16-03562-f013:**
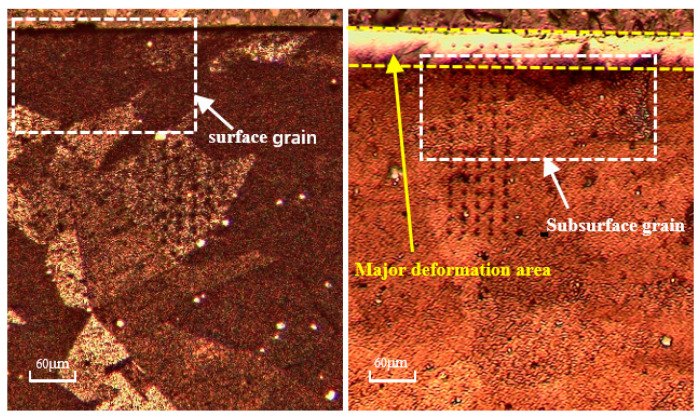
Microscope photos of machined surface and subsurface.

**Figure 14 materials-16-03562-f014:**
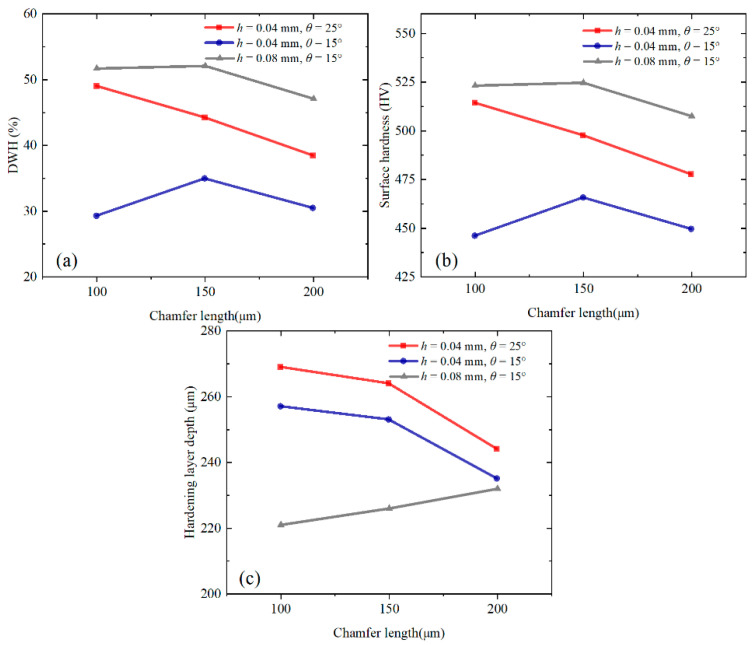
Variation trends of DWH (**a**), surface hardness (**b**), and hardness layer depth (**c**).

**Table 1 materials-16-03562-t001:** J–C material constitutive model parameters of Inconel 718 [27].

Parameter	*A* (MPa)	*B* (MPa)	*C*	*n*	*m*
Value	1290	895	0.016	0.526	1.55

**Table 2 materials-16-03562-t002:** DDB model parameters of Inconel 718 [35].

**Parameter**	** *α** **	** *β** **	** *B* **	***b* (*mm*)**	${ƒ}_{0}$	${ƒ}_{\infty }$
Value	0.08	0.04	14,900	2.5×10^7^	0.25	0.06
**Parameter**	** *K* **	** *M* **	${\dot{\gamma }}_{0}^{r}$	${\tilde{\gamma }}^{r}$	$\rho_{w}^{t=0}$ **(*mm^−2^*)**	$\rho_{c}^{t=0}$ **(*mm^−2^*)**
Value	119 or 769	3.06	1.0 × 10^5^	3.2	2.5 × 10^7^	5.0 × 10^7^

**Table 3 materials-16-03562-t003:** Cutting parameters for Inconel 718 machining process.

Test	Cutting Speed: *v_c_* (m/min)	Uncut Chip Thickness: *h* (mm)	Chamfer Angle: *θ* (°)	Chamfer Length: *L* (μm)
1	50	0.04	25	100
2	50	0.04	25	150
3	50	0.04	25	200
4	50	0.04	15	100
5	50	0.04	15	150
6	50	0.04	15	200
7	50	0.08	15	100
8	50	0.08	15	150
9	50	0.08	15	200

## Data Availability

Not applicable.

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
