# Peer review of "Simulation of Work Hardening in Machining Inconel 718 with Multiscale Grain Size"

_materials, 2023, doi:10.3390/ma16093562_

Round 1

Reviewer 1 Report

Find the review in the attached pdf file.

Reviewer 2 Report

The present work determines the work hardening produced by turning using chamfered inserts using two ways: computational simulation and experimental tasks.

The work is very interesting, complete, it is very well conducted and it yields consistent results between the modeling and the experimental values.

Despite this, the authors must improve and respond to some specific aspects:

1) Although the authors seem to have analyzed 34 references, their list does not appear in the paper

2) To facilitate the reading of the paper, a complex work in a variety of activities such as this requires for its better understanding a single flowchart that includes both the simulation and experimental tasks differentiated by grain size and that contains all the variables considered and the answers obtained.

3) Regardless of the grain size considered, ¿why did you choose such a small range of cutting depths (0.04 to 0.1 mm)?

4) It is necessary to indicate in schematic form the shape and dimensions of the turned specimen.

5) What causes the change in cutting speed and cutting depth when testing the material of small and large grain size?

6) Why are figures 9 and 12 d repeated?

7) In the conclusions it is necessary to point out 3 or 4 relevant aspects of the work.

Reviewer 3 Report

This manuscript describes the simulation of the work hardening of Inconel 718 based on recrystallization and dislocation density models for different grain sizes. The text is well-written. However, it is unacceptable to submit a version with revisions that are not accepted. The authors need to submit a revised version with all changes accepted. In addition, the overall English of the manuscript needs to be improved. 

Reviewer 4 Report

For their work, the authors propose a numerical model to estimate the work hardening layer for a deeper understanding of this behavior, employing both recrystallization–based and dislocation–based models to cover workpieces with multiscale grain size. They reported that the validation of the proposed model by performed with both literature and experimental data for Inconel 718 with small or large grain sizes. The work done is very interesting and complete, however, there are some observations I would like to make about it

To authors

1.      I received a non-final version of the manuscript, as it has many corrective remarks in the text, which causes some confusion.

2.      Figure 2 is not mentioned in the manuscript.

3.      In equations 4 and 5 what values were assigned to the constants b, m, C0, and C1 and from where they were obtained.

4.      If they reported the standard deviation of the experimental measurements in Figure 6 surely the % error would be even less significant.

5.      There is not sequence in the numbering of the tables. (i.e. Table 3 appears before Table 2, and there are two Tables 3).

6.      It is not clear from the experimental phase how the measurements of dislocation density, grain size and hardness were carried out.

7.      A thorough revision of the manuscript is suggested, taking care of numbering, punctuation and interlinear spacing, since the text of the manuscript appears immediately after the caption of the figures and it is not known where one begins and the other ends.

Round 2

Reviewer 1 Report

Please find my comments in the attached pdf file to your paper.

Reviewer 2 Report

The paper has improved significantly.

Reviewer 3 Report

The comments has been addressed and can be accepted in the current form for publication.